# Solvation, Cancer Cell Photoinactivation and the Interaction of Chlorin Photosensitizers with a Potential Passive Carrier Non-Ionic Surfactant Tween 80

**DOI:** 10.3390/ijms23105294

**Published:** 2022-05-10

**Authors:** Andrey V. Kustov, Philipp K. Morshnev, Natal’ya V. Kukushkina, Nataliya L. Smirnova, Dmitry B. Berezin, Dmitry R. Karimov, Olga V. Shukhto, Tatyana V. Kustova, Dmitry V. Belykh, Marina V. Mal’shakova, Vladimir P. Zorin, Tatyana E. Zorina

**Affiliations:** 1United Physicochemical Centre of Solutions, G.A. Krestov Institute of Solution Chemistry, Russian Academy of Sciences (ISC RAS), 153045 Ivanovo, Russia; morshnevphilipp@gmail.com (P.K.M.); nataliakukushkina05.1998@mail.ru (N.V.K.); smir141973@mail.ru (N.L.S.); 2Institute of Macroheterocyclic Compounds, Ivanovo State University of Chemistry and Technology (ISUCT), 153012 Ivanovo, Russia; dmitriy.karimov@list.ru (D.R.K.); shukhto@isuct.ru (O.V.S.); melenchuktv@mail.ru (T.V.K.); 3Institute of Chemistry of the Komi Science Centre of the Ural Branch of Russian Academy of Sciences (ICKSC UB RAS), 167000 Syktyvkar, Russia; belykh-dv@mail.ru (D.V.B.); berezin1@rambler.ru (M.V.M.); 4Department of Biophysics, Belarussian State University (BSU), 220030 Minsk, Belarus; vpzorin@mail.ru (V.P.Z.); zorinate@mail.ru (T.E.Z.)

**Keywords:** photodynamic therapy, chlorin photosensitizers, 1-octanol/phosphate saline buffer partition, solvation, singlet oxygen generation, photoinactivation, interaction with Tween 80

## Abstract

Cancer and drug-resistant superinfections are common and serious problems afflicting millions worldwide. Photodynamic therapy (PDT) is a successful and clinically approved modality used for the management of many neoplastic and nonmalignant diseases. The combination of the light-activated molecules, so-called photosensitizers (PSs), with an appropriate carrier, is proved to enhance PDT efficacy both in vitro and in vivo. In this paper, we focus on the solvation of several potential chlorin PSs in the 1-octanol/phosphate saline buffer biphasic system, their interaction with non-ionic surfactant Tween 80 and photoinactivation of cancer cells. The chlorin conjugates containing *d**-*galactose and *l*-arginine fragments are found to have a much stronger affinity towards a lipid-like environment compared to ionic chlorins and form molecular complexes with Tween 80 micelles in water with two modes of binding. The charged macrocyclic PSs are located in the periphery of surfactant micelles near hydrophilic head groups, whereas the *d*-galactose and *l*-arginine conjugates are deeper incorporated into the micelle structure occupying positions around the first carbon atoms of the hydrophobic surfactant residue. Our results indicate that both PSs have a pronounced affinity toward the lipid-like environment, leading to their preferential binding to low-density lipoproteins. This and the conjugation of chlorin e_6_ with the tumor-targeting molecules are found to enhance their accumulation in cancer cells and PDT efficacy.

## 1. Introduction

Photodynamic therapy (PDT) has proved to be an efficient and safe treatment for many socially significant diseases, such as superficial malignancies of different etiology and localized antibiotic-resistant microbial infections [1,2,3,4,5,6,7,8,9,10,11,12,13,14]. PDT is considered an alternative for, or the important part of, combined treatment, including traditional surgical intervention, chemo- or antibiotic therapy [2,3,4,5,6,7,8,12,13,14,15,16]. This unique technique can be successfully applied both for fluorescent-guide resection of tumor tissue and local photoinactivation of a pathological process. The latter often induces a PDT-mediated enhancement of antitumor immunity due to activation of macrophages and dendritic cells [1,14,17]. The important advantage of PDT compared to other methods of treating localized infections is that microorganisms do not develop resistance to repeated PDT sessions [5,6]. In contrast, cancer cells are able to exploit several cytoprotective mechanisms to avoid oxidative stress induced by the PDT treatment [1,17].

Despite the great success of clinical PDT during the last decades [1,2,3,12,13,14], there are several limitations of the second-generation PSs that are currently in use. These are in the insufficient selectivity of PS accumulation in tumor tissue, aggregation of the porphyrin- and chlorin-type PSs at therapeutic concentrations, the limited depth of red light penetration through tissue for treating deeper located malignancies, weak photoinactivation of Gram-negative microorganisms, etc. [1,2,14]. The conjugation of PS with various tumor-targeting molecules or the application of an appropriate passive carrier appears to enhance PDT efficacy [18,19,20,21,22]. In fact, the selectivity of PS accumulation in a malignant tissue can be enhanced by taking into account the features of the metabolism of tumor cells. It is known that these cells are prone to accumulate a variety of certain substances necessary for their rapid growth [1,23]. These compounds are actively involved in cell metabolism, which is confirmed by the high efficiency of positron emission tomography and fluorescence diagnostics of cancer with appropriate agents containing amino acids, vitamins or sugars [7,21,24,25]. Thus, one can expect that the application of chlorin e_6_ conjugates with the active fragments of *l*-arginine or *d*-galactose should increase the selectivity of PS accumulation in tumors and enhance PDT efficacy. In this paper, we compare solvation, the PS-carrier interaction and tumor cell photoinactivation for several chlorin PSs containing neutral, zwitterionic or charged groups (see comps. **1**–**4** in Figure 1), which seem to be appropriate candidates for antitumor or antimicrobial PDT.

## 2. Results and Discussion

### 2.1. Partition Coefficients and PS Solvation in a Lipid-like and Aqueous Environment

The human body can be viewed as a series of lipid-like barriers dividing aqueous filled compartments [26]. Thus, the affinity of PS molecules towards a lipid-like environment strongly influences their binding to serum proteins and passive transport through cellular membranes. The simplest model of the inner core of a lipid membrane is liquid 1-octanol (OctOH) [26,27,28], which is used here to study PS partition between two immiscible phases modeling an aqueous and lipid-like environment.

Partition coefficients (*P*) between OctOH and phosphate saline buffer (PSB) were determined with the method of isothermal saturation [28,29]:*P* = *m*_OctOH_/*m*_aq_, (1)
where *m*_OctOH_ and *m*_aq_ are solute equilibrium molalities in OctOH and PSB, respectively. The equilibrium concentration of PSs was analyzed spectrophotometrically using previously obtained calibration plots in 1-octanol or PSB. The corresponding partition coefficients of neutral and charged PSs between OctOH and PSB at 298–318 K are compared in Table 1. The thermodynamic functions of solute transfer were calculated according to Equation (2):(2)$$R\ln P=-\frac{\Delta_{t}G{\;}^{0}}{298.15}+\Delta_{t}H{\;}^{0}\times \left[\frac{1}{298.15}-\frac{1}{T}\right],$$
where $\Delta_{t}G{\;}^{0}=-R\ln P$ is the standard free energy of PS transfer from PSB to OctOH, 298.15 is the reference temperature and Δ_t_*H*^0^ is the standard enthalpy of transfer at the reference temperature, respectively. These values are also given in Table 1.

We see that for trisodium salt of chlorin e_6_ (comp. **1**) and tricationic chlorin (comp. **4**), the *P* values were almost temperature-independent and much smaller than those for the chlorin conjugates with the fragments of *d*-galactose (comp. **2**) or *l*-arginine (comp. **3**). The most hydrophilic comp. **4** was equally distributed between PSB and 1-octanol in the whole temperature range studied, whereas the equilibrium molality of chlorin e_6_ trisodium salt (comp. **1**) was about twice as large in a lipid-like phase. The partition coefficients for neutral (comp. **2**) and zwitterionic (comp. **3**) PSs were much larger. These solutes were slightly soluble in water and localized mainly in a lipid-like phase. This intuitively expected result suggests that ionic groups of charged PSs provide the pronounced enhancement of macrocycle affinity towards a water-like pool and, simultaneously, weaken its transfer through lipid membranes both at room and elevated temperatures. It is known [24,30,31] that chlorin e_6_ trisodium salt (*P*~2) binds to serum albumin in a vascular system, whereas more hydrophobic PSs (*P* ≥ 10) are mainly transported by high- and low-density lipoproteins. Thus, comps. **2**, **3,** containing the active fragments of *d*-galactose or *l*-arginine (*P* > 15), should be mainly bound to blood lipoproteins. Since many types of tumor cells have appropriate receptors to amino acids or sugars and accumulate lipoproteins, we suppose that comps. **2**, **3** must be selectively accumulated in tumor tissue.

Table 1 gives the thermodynamic quantities of PS transfer from aqueous phosphate saline buffer to OctOH. We see that the free energy of transfer was negative for all cases except the most hydrophilic tricationic chlorin, where it approached zero. The enthalpy term was positive, indicating that solvation of the PSs was more exothermic in PSB compared to OctOH due to attractive interactions between solute polar/charged groups and water molecules. In contrast, the entropic term was negative and favored solute transfer to a lipid-like phase in the whole temperature range studied. This solute behavior is typical for the water-soluble amphiphilic species with a large hydrophobic residue [28].

### 2.2. Aggregation PS at Therapeutic Concentrations

PS aggregation in water seems to be a serious problem in clinical PDT because *π*-stacking of PS molecules may significantly reduce the efficiency of ROS generation [7,10,14,19], drug bioavailability and its affinity to cell membranes [6,14,24]. In principle, PS aggregation can be prevented by using appropriate passive or active delivery systems, such as polymer or micellar surfactants, liposomes, nanoparticles or vector carriers [18,19,20,32].

The well-soluble ionic PSs (see comps. **1**, **4** in Figure 1) were found to be aggregated at therapeutic concentrations. According to our DLS study, comp. **4** formed large aggregates in water with a hydrodynamic diameter of 150–200 nm at the concentration of 10^−3^ mol kg^−1^ or higher [32]. In contrast, chlorin e_6_ trisodium salt aggregated at much larger concentrations. It is rather surprising because according to the *P* values in Table 1, comp. **1** is more hydrophobic than tricationic chlorin. Hence, although both PSs have an identical number of charged groups, their aggregation behavior differs. Apparently, this indicates that both the number and type of charged groups, as well as their relative location in the PS molecule, affect the aggregation process. As for hydrophobic comps. **2**, **3**, it is expected that they must form such aggregates at much lower concentrations. This and the low solubility mentioned above make it compulsory to use appropriate carriers for both chlorins.

### 2.3. Interaction with Tween 80

Tween 80 is a biocompatible non-ionic micellar surfactant (CMC = 1.2 × 10^–5^ M) that is widely used in pharmacology as an effective agent for solubilization and passive transport of many drugs [33,34]. Our recent model calculations have indicated [35,36,37,38] that Tween 80 forms relatively stable complexes with several PSs, and most revealed two modes of PS-surfactant binding. In this study, we applied this approach [35,36,37,38] toward the interaction between Tween 80 and chlorin PSs, as shown in Figure 1. The model parameters, *viz*. lg *K*_b_ and *n*, given in Table 2 were recovered from the experimental titration curves by fitting them to Equation (3):lg[(*A* − *A*_0_)/(1 − (*A* − *A*_0_))] = lg(*K*_b_) + *n×*lg[*m*_T_^m^ – *n×m*_PS_·(*A* − *A*_0_)/(*A*_max_ − *A*_0_)],(3)
where *m*_PS_ is the PS concentration equaling to ~4–7 10^−6^ mol kg^−1^; *m*_T_^m^ = *m*_T_ − CMC is the concentration of aggregated Tween 80, which is evaluated as the difference between its analytical concentration and critical micellar concentration; *n* is the number of surfactant molecules in close contact with a PS in a micelle; *A*_0_ and *A*_max_ are the optical densities of a PS solution in pure water and in an aqueous solution of Tween 80, where the experimental curve of *A* vs. *m*_T_ reaches the plateau.

Table 2 shows that all the solutes demonstrated two modes of PS-surfactant binding. Since the Tween 80 aggregation number in water is 60 [34,35,36], there is a deficit of micelles in a PS solution at low surfactant concentrations. Here, PSs interacted on average with 0.5–1.5 surfactant molecules, indicating that there are PS-PS contacts on the micelle surface [36]. The binding constant was the largest for hydrophobic zwitterionic comp. **3**, whereas the most hydrophilic tricationic chlorin (comp. **4**) had the smallest lg *K*_b_ value equal to 1.4.

For larger surfactant concentrations, the second mode of binding with the larger lg *K*_b_ and *n* values was detected. The increase in the surfactant concentration led to the rapid growth of the micelle number in a solution. This fact and accommodation of bulky PS molecules in the micelle structure induced the formation of larger aggregates, where the macrocycles interact with the larger number of surfactant molecules (see Table 2).

Fluorescence quenching in a PS aqueous solution containing Tween 80 may shed additional light on the most probable location of chlorin molecules in the carrier structure. The PS quenching curves by potassium iodide are well-described by the famous Stern–Volmer equation:*F*_0_/*F* = 1 + *K*_SV_
*×* [I^−^],(4)
where *F*_0_ and *F* are intensity of PS fluorescence in an aqueous solution of Tween 80 without the quencher and KI solutions, respectively; [I^−^] is the molal concentration of the quencher; *K*_SV_ is the quenching constant.

The results given in Table 2 show that for the surfactant/PS molar ratio of 60, the corresponding quenching constants are large enough. The *K*_SV_ value is the largest for well-hydrated tricationic chlorin, indicating its location near hydrophilic head groups, where iodide ions can easily penetrate and bind to the macrocycle. This superficial location of comp. **4** is in good agreement with the high affinity of this PS to the water-like compartment (see Table 1). Chlorin e_6_ trisodium salt (comp. **1**) has a smaller *K*_SV_ value, strongly decreasing at larger Tween 80 concentrations. This quantity is nearly identical to the *K*_SV_ value of the most hydrophobic comp. **2**. We have mentioned above that the interaction between labile Tween 80 micelles loaded by PSs enhances PS-surfactant binding. Table 2 shows that this finding was independently confirmed by the quenching experiments. It is apparent that non-ionic chlorin PSs (comps. **2**, **3**) occupy intermediate positions in the palisade layer of Tween 80 micelles near the first carbon atoms of the hydrophobic residue, where well-hydrated iodide-ions are not able to penetrate to induce efficient quenching of solute fluorescence.

### 2.4. Spectral Characteristics and Generation of Singlet Oxygen

The absorption spectra of comps. **1**–**4** were found to be typical for the chlorin-type macrocycles [28] and quite similar both in ethanol and an aqueous solution of Tween 80 (see the Appendix A). The 18 π-electron chromophore system is responsible for the appearance of the intensive Soret (B-) band near 400 nm (lg*ε* > 5) and less pronounced Q-bands between 500–670 nm. The most intensive bathochromic Q_x(0–0)_-band at 660 nm (lg*ε* = 4.5–4.7) induced by π-π*-electron transfer is within the so-called optical window of tissue [5,6,7,13,14,15,16]. The fluorescence emission spectra of comps. **1**–**4** demonstrate the intensive band between 660 and 670 nm accompanied by a moderate Stokes shift of 5–10 nm. The absorption and fluorescence spectra for comps. **2**, **3** are illustrated in Appendix A, whereas the other ones are given elsewhere [13,28].

The quantum yield of singlet oxygen (*Φ*_Δ_) is one of the key quantities of any PS designed for PDT [14,15,16]. In this paper, the *Φ*_Δ_ values were determined by the indirect chemical method [16] using the following equation:(5)$$\Phi_{\Delta }{\;}_{\mathrm{PS}}=\frac{k_{d\;\mathrm{PS}}}{k_{d\;\mathrm{St}}}\;\frac{IP_{\mathrm{St}}}{IP_{\mathrm{PS}}}\Phi_{\Delta }{\;}_{\mathrm{St}}$$
where the St and PS symbols refer to the standard and PS solutions, respectively; *k*_d_ is the rate-of-degradation constant; the *IP* value is estimated numerically by:(6)$$IP=\int_{\lambda_{1}}^{\lambda_{2}}I_{0}\left(\lambda \right){\left(1-10^{-A}\right)}_{\;}d\lambda$$
where *λ*_1_ and *λ*_2_ are the initial and final wavelengths of the region where the spectra of a light source and a PS overlap; *I*_0_ (*λ*) is the intensity of the source as a function of *λ* and *A* is the optical density of a solution. Table 3 shows that both novel PSs efficiently generated singlet oxygen species, the *Φ*_Δ_ values being slightly larger than those for charged chlorins.

### 2.5. Photodynamic Activity and Photosensitizers Accumulation in Cancer Cells In Vitro

It is known [14,17,18] that the affinity of PS molecules towards specific receptors on a cancer cell surface or at least toward a lipid-like compartment enhances PS accumulation in tumor tissue and leads to an increase in PDT efficacy. Our results given in Table 3 support this point of view and indicate that the most hydrophobic PSs, *viz*. comps. **2** and **3** (*P* > 15), are selectively accumulated in K-562 cells compared to hydrophilic chlorin e_6_ trisodium salt (comp. **1**). This effect is well correlated with higher phototoxicity of these PSs at both light doses of 0.44 and 0.66 J cm^−2^ (see Table 3). Thus, comps. **2**, **3** demonstrate thrice as large PDT efficacy towards cancer cells compared to clinically approved chlorin e_6_ trisodium salt.

## 3. Materials and Methods

### 3.1. Photosensitizers and Other Chemicals

Chlorin e_6_ trisodium salt (comp. **1**, “Fotoran e_6_”) was purchased from “RANFARMA” company (Russia) as a solid powder mixed with polyvinylpyrrolidone (PVP). The drug was reprecipitated from an aqueous solution at pH ≈ 6.0, centrifuged and washed several times to remove the residual surfactant. Then, it was dissolved in water at pH ≈ 8.0. After that, the solvent was evaporated to obtain solid comp. **1** (for more details, see the Appendix A).

The conjugate of chlorin e_6_ with *d*-galactose (comp. **2**) was synthesized according to the recently published procedure [39] by means of the interaction of the activated 17(3)-carboxylic group in a chlorin molecule with the protected *d*-galactose derivative, followed by hydrolysis of the protecting groups (see Scheme S1 in the Appendix A).

The novel conjugate with *L*-arginine (comp. **3**) was synthesized from methylpheophorbide *a* [40] using a carboxylic group activation reaction through the formation of succinimide ester according to Schemes S2-S5 shown in the Appendix A.

Tricationic chlorin e_6_ (comp. **4**) was obtained by chemical functionalization of methylpheophorbide *a,* as described earlier [28] (see Scheme S6 in the Appendix A).

The detailed description of synthesis and spectral identification of comps. **1**–**4** are given in the Appendix A). The ^1^H NMR spectra of the PSs studied are listed below.

Comp. **1**
^1^H NMR (500 MHz, DMSO d_6_, *J*, Hz), *δ*, ppm: 9.75 (s, 1H, H-10); 9.60 (s, 1H, H-5); 9.08 (s, 1H, H-20); 8.33 (m, 1H, H-3(1)); 6.43 (d, *J* = 16.0, 1H, H-3(2) (*trans*)); 6.13 (d, *J* = 10.5, 1H, H-3(2) (*cis*)); 5.66 (br. s., 2H, CH_2_-15(1)); 4.54 (m, 2H, H-18, H-17); the signals from 4 to 1 ppm seem to be covered by residual PVP; −2.06 (br. s., 1H, 21-NH); −2.62 (br. s., 1H, 23-NH).

Comp. **2**. ^1^H NMR (300 MHz, DMSO d_6_, *J*, Hz), *δ*, ppm: 9.81 (s, 1H, H-10); 9.77 (s, 1H, H-5); 9.15 (s, 1H, H-20); 9.07 (m, 1H, NH-13(1) (amide)); 8.33 (dd, *J* = 11.6, 17.9, 1H, H-3(1)); 6.46 (d, *J* = 17.4, 1H, H-3(2) (*trans*)); 6.19 (d, *J* = 12.0, 1H, H-3(2) (*cis*)); 5.54 (d, *J* = 19.2, 1H), 5.37 (m, 1H) (CH_2_-15(1)); 4.89 (d, *J* = 12.3, 1H, H-1^a^); 4.66 (m, 1H, H-18); 4.50 (d, *J* = 9.3, 1H, H-17); 4.65 (d, *J* = 7.2, 2H,), 4.19-3.89 (m, 4H) (H-2^a^, H-3^a^, H-4^a^, H-5^a^, CH_2_-6^a^); 3.84 (q, *J* = 7.3, 2H, CH_2_-8(1)); 3.71 (s, 3H, CH_3_-15(3)); 3.54 (s, 3H, CH_3_-12(1)); 3.51 (s, 3H, CH_3_-2(1)); 3.33 (s, 3H, CH_3_-7(1)); 3.67-3.42 (m, 4H, OH-1^a^, OH-2^a^, OH-3^a^, OH-4^a^); 3.13 (d, *J* = 3.9, 3H, CH_3_-13(2)); 2.80–2.55 (m, 2H), 2.26–2.04 (m, 2H) (CH_2_-17(1), CH_2_-17(2)); 1.77–1.59 (m, 6H, CH_3_-18(1), CH_3_-8(2)); −1.82 (br. s., 1H, 21-NH); −2.09 (br. s., 1H, 23-NH).

Comp. **3**. ^1^H NMR (500 MHz, DMSO d_6_, *J*, Hz), *δ*, ppm: 9.77 (s, 1H, 10-H); 9.73 (s, 1H, 5-H); 9.38 (br. s., 1H, 17(3)-NH); 9.07 (s, 1H, 20-H); 9.03 (d, *J* = 4.0 Hz, 1H, 13-NCH_3_); 8.26 (dd, *J* = 11.8, 17.9 Hz, 1H, 3(1)-H); 6.85–8.05 (br. m., 5H, NH, guanidine group); 6.40 (d, *J* = 17.8 Hz, 1H, 3(2)-H-*trans*); 6.15 (d, *J* = 11.7 Hz, 1H, 3(2)-H-*cis*); 5.52 (d, *J* = 18.8 Hz, 1H, 15(1)-CH); 5.32 (d, *J* = 18.5 Hz, 1H, 15(1)-CH); 4.62 (dd, *J* = 6.9, 14.2 Hz, 1 H, 18-H); 4.38 (d, *J* = 10.35 Hz, 1H, 17-H); 4.02 (dd, *J* = 6.4, 13.2 Hz, 1H, C1’-CH); 3.83 (q, *J* = 6.8 Hz, 2H, 8(1)-CH_2_); 3.68 (s, 3H, 15(3)-CH_3_); 3.49 (s, 3H, 12(1)-CH_3_); 3.44 (s, 3H, 2(1)-CH_3_); 3.32 (s, 3H, 7(1)-CH_3_); 3.11 (d, *J* = 4.4 Hz, 3H, 13(1)-NCH_3_); 3.00 (m, 2 H, C4’-CH_2_); 2.55 (s, 1H); 17(1)-CH_2_, 17(2)-CH_2_): 2.12, 2.19 (2m, 1H each), 2.57–2.66 (m, 2H); 1.68 (t, *J* = 7.5 Hz, 3H, 8(2)-CH_3_); 1.63 (d, *J* = 7.2 Hz, 3H, 18(1)-CH_3_); 1.56 (m, 2H, C2’-CH_2_); 1.38, 1.44 (2m, 1H each, C3’-CH_2_); −1.84 (s, 1H, 21-NH); −2.12 (s, 1H, 23-NH).

Comp. **4**. ^1^H NMR (500 MHz, DMSO d_6_*, J*, Hz), *δ*, ppm: 9.87 (2 H, s, 10-H, 5-H); 8.59–8.46 (br. m., 1H, 13-CONHCH_2_CH_2_N^+^(Me)_3_I^−^); 9.17 (s, 1H, 20-H); 7.39–7.22 (br. m., 1H, 3-C(CH_2_N^+^(Me)_3_I-)=CHCH_2_N^+^(Me)_3_I^−^); 5.50 (d, *J* = 19.2 Hz, 1H, 15-CH_A_H_B_CO_2_Me); 5.35 (d, *J* = 19.2 Hz, 1H, 15-CH_A_H_B_CO_2_Me); 4.67 (q, *J* = 7.3 Hz, 1H, 18-H); 4.48 (br. d., *J* = 8.3 Hz, 1H, 17-H); 4.13–3.99 (m, 2H, 13-CONHCH_2_CH_2_N(Me)_2_); 3.96-3.82 (m, 2H, 8-CH_2_Me); 3.88 (s, 3H, 15-CH_2_CO_2_Me); 3.76 (s, 3H, 17-CH_2_CH_2_CO_2_Me); 3.59 (s, 6H, 12-Me, 7-CH_3_); 3.57 (s, 3H, 2-Me); 3.39 (s, 18H, 3-C(CH_2_N^+^(Me)_3_I^−^)=CHCH_2_N^+^(Me)_3_I^−^)); 3.16 (s, 9H, 13-CONHCH_2_CH_2_N^+^(Me)_3_); 3-C(CH_2_N^+^(Me)_3_I^−^)=CHCH_2_N^+^(Me)_3_I^−^: 3.05-2.88 (m, 2H), 2.83-2.65 (2H, m); 2.34-2.24 (m, 2H, 13-CONHCH_2_CH_2_N^+^(Me)_3_I^−^), 2.21-1.89 (m, 4H, 17-CH_2_CH_2_CO_2_Me), 1.72 (t, 3H, *J* = 7.3 Hz, 8-CH_2_Me), 1.67 (d, *J* = 7.2 Hz, 3H, 18-Me), −1.82 (br. s., 1H, 23-NH), −2.09 (br. s., 1H, 21-NH). All PS solutions were prepared by weight, filtered with the 0.22 μm cellulose filter and stored in a dark cool place for several days.

1-Octanol (Panreac, >98%) was dried with 4 Å molecular sieves and distilled under reduced pressure at 355 K. Phosphate saline buffer with pH~7.4 (Agat-med, for biochemical laboratories) was prepared by dissolving pure solid forms in one liter of freshly bidistilled water [8,28]. Tween 80 (Panreac, pharma grade), 1,3-diphenylisobenzofuran (DPBF) (J&K Scientific Gmbh, Germany, >97%), ethanol (Sigma-Aldrich, St. Louis, MO, USA, ≥99%), RPMI-1640 growth medium (Sigma-Aldrich), fetal calf serum (FCS, Sigma-Aldrich) and potassium iodide (Sigma-Aldrich, ≥99%) were used as supplied.

### 3.2. Thermodynamic and Stectroscopic Studies

Partition coefficients between OctOH and phosphate saline buffer were determined by the method of isothermal saturation described several times [8,28,29]. Weighed amounts of a sensitizer solution in PSB (comps. **1**, **4**) or OctOH (comps. **2**, **3**) with the initial concentration of 8–15 μmol kg^−1^ and the volume ratio of 50:50 were placed into a cell and intensively stirred with a magnetic stirrer, usually for 24 h. The temperature instability in the cell during the experiment was within ±0.1 K. When equilibrium was reached, the stirrer was switched off to achieve phase separation. After four–five hours, three milliliters of an aqueous or lipid-like fraction were carefully taken up with a stainless steel needle and analyzed spectrophotometrically using previously obtained calibration plots.

UV-Vis spectra were recorded with the D8 spectrophotometer (Drawell, China) at room temperature. Fluorescence spectra were registered with the Solar CM 2203 spectrofluorimeter at 298 K (Belarus). PS solutions were prepared by dissolving an appropriate amount of a macrocycle in pure water, OctOH or an aqueous Tween 80 solution with the Sonopuls ultrasonic homogenizer (Bandelin electronic GmbH & Co., Germany). The solute concentration was 4–7·10^−6^ mol kg^−1^ and 1·10^−6^ mol kg^−1^ for spectrophotometric and fluorescence experiments, respectively.

For spectrophotometric measurements, each solution was placed into the spectrophotometer cell 10–15 min before to record UV-Vis spectra. For comps. **1**–**3,** the chlorin–surfactant interaction was analyzed at 665–670 nm, whereas for comp. **4,** the wavelength was chosen to be 395 nm.

The study of fluorescence quenching was carried out by titration of chlorin aqueous solutions containing Tween 80 by aqueous solutions of KI as an appropriate ionic quencher [36]. The surfactant/PS molar ratio was equal to 60 or 200 and kept constant during titration. The excitation wavelength equaled 505 nm, and emission was registered in the range of 550–750 nm. The experimental results are given in the Appendix A (see Appendix A).

The dynamic light scattering study (DLS) of PS solutions was performed with the Zetasizer Nano ZS apparatus ZEN 3600 (Malvern Instruments, Great Britain) equipped with a laser with λ = 633 nm and a non-invasive backscatter technology [32]. PS solutions were stored in glass vessels protected from sunlight. The experiments were performed a couple of days after the preparation of solutions, and the scattering intensity distribution was analyzed.

The quantum yield of singlet oxygen was indirectly determined by the well-known chemical method in liquid OctOH [16]. The monocationic chlorin (13(1)-N-(2-N′N′N′-trimethylammonioethyl iodide)amide chlorin e_6_ 15(2),17(3)-dimethyl ester) with the known *Φ*_Δ_ value equaling 0.65 [28] and DPBF were used as the standard PS and the selective singlet oxygen trap, respectively. Two quartz cuvettes contained a PS + DPBF solution in OctOH, and a standard PS + DPBF solution was irradiated with the LED diode panel (BIC, Belarus’) [14,16], emitting between 590 and 720 nm with a maximum at 662 nm. The red light dose of 0.18 J cm^−2^ was delivered every minute. DPBF degradation was monitored spectrophotometrically in both cuvettes and each irradiation session contained six–eight measurements. Then, the rate-of-degradation constants *k*_d_ (see Equation (5)) were evaluated in terms of the first-order exponential decay. All photo-bleaching experiments were repeated from 5 to 7 times.

### 3.3. Biological Assays

Light toxicity of PSs **1**–**3** towards tumor cell lines was studied with the K-562 myeloid leukemia cell model. Malignant cells from the collection of the Belorussian Research Center of Pediatric Oncology, Hematology and Immunology (Minsk, Belarus’) were cultured in RPMI-1640 growth medium containing 5% *v*/*v* fetal calf serum. All manipulations were nearly identical to those described before [41]. Briefly, comps. **1**–**3** dissolved in ethanol were added to RPMI-1640 medium containing 5% FCS and incubated for 1 h at 37 °C. Then, an appropriate amount of this mixture was added to the growth medium containing 10^6^ cells to reach the final PS concentration of 10 μM. After PS accumulation, the cells were washed several times by a growth medium.

PSs accumulation in K-562 cells was studied with the TCS SPE laser scanning confocal fluorescence microscope (Leica, Germany) equipped with an immersion (63×) magnification objective. An argon laser with λ = 488 nm was used as an appropriate fluorescence excitation source. Fluorescence emission was registered from 620 to 700 nm.

For the analysis of PS light toxicity, K-562 cells were incubated with an appropriate PS and then illuminated by red light (TLM-660-0.5 diode laser, Belarus’) for 20–30 s with a light dose of 0.44 or 0.66 J cm^−2^ (Table 3). After irradiation, the cells were incubated again for 3 h. The percentage of dead cells was determined by means of fluorescence intensity measurements using the FC 500 cytometer equipped with the CXP statistical software package (Beckman Coulter, Brea, CA, USA). The excitation and emission wavelengths were 488 nm and 520 nm, respectively. All experiments mentioned above were repeated 3–5 times.

## 4. Conclusions

In summary, it is reasonable to make a brief description of the results obtained both in this paper and several related studies, which illuminate some of the important features contributing to the PS behavior in vivo. First, it should be mentioned that all four PSs shown in Figure 1 had a large enough quantum yield of singlet oxygen in a lipid-like phase reaching 0.55–0.75 [13,16,31]. This indicates that these species may have some potential in PDT. However, a large quantum yield is less important for the clinical utilization of any PS compared to its biodistribution in the human body [1,14]. Second, the PSs studied demonstrate different affinity towards a lipid-like compartment. The ionic species are much more hydrophilic, which is in obvious agreement with chemical intuition. In contrast, neutral and zwitterionic chlorins are preferentially located in a lipid-like phase; this tendency slightly enhances with temperature. Thus, these PSs should demonstrate the pronounced selectivity of accumulation in malignant cells due to their affinity to a lipid-like compartment and the existence of tumor-targeted fragments on the periphery of a PS molecule. This finding is supported by the cell experiments presented in Table 3. Third, the study of the PS-interaction with a potential carrier Tween 80 clearly indicates that macrocycles form stable enough molecular complexes with surfactant micelles with two modes of binding. The ionic PSs, and, especially, cationic chlorins, are located on the periphery of surfactant micelles near hydrophilic head groups. In contrast, the hydrophobic chlorin conjugates with *d*-galactose and *l*-arginine are incorporated deeper into the micelle structure and seem to be localized in the palisade layer around the first carbon atoms of the hydrophobic residue. This PS behavior is well correlated with the partition coefficients between aqueous and lipid-like phases. These results may have implications with regard to the development of a potential delivery system based on the Tween 80 platform, where the strength of binding may be regulated both by the surfactant concentration and the PS structure.

## Figures and Tables

**Figure 1 ijms-23-05294-f001:**
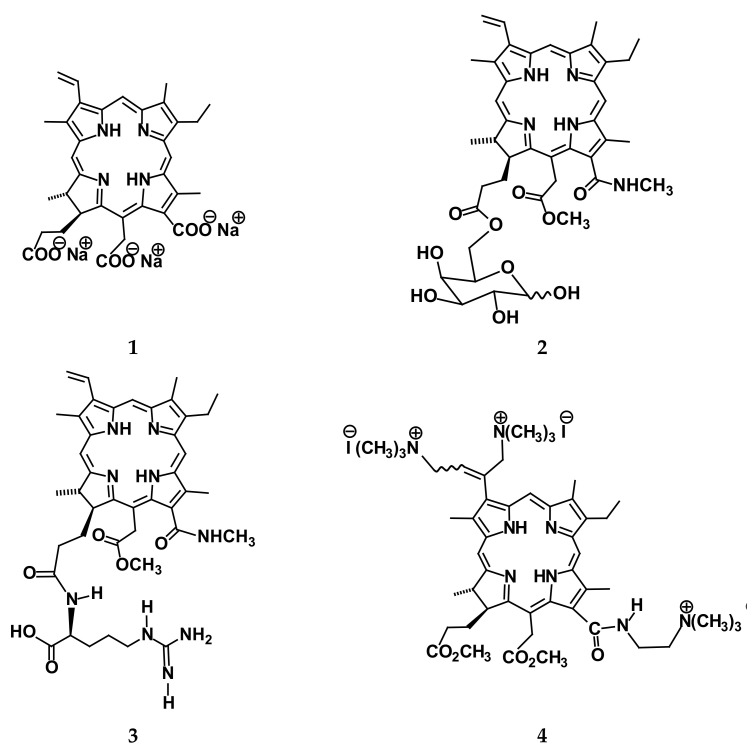
Molecular structures of the PSs studied: **1**—chlorin e_6_ trisodium salt (comp. **1**); **2**—chlorin e_6_ 13(1)-N-methylamide-15(2)-methyl ester-17(3)-O-6′-galactopyranosyl ester (comp. **2**); **3**—chlorin e_6_ 13(1)-N-methylamide-15(2)-methyl ester-17(3)-[N-1′-(1′-carboxy-4′-guanidylbutyl) amide] (comp. **3**); **4**—chlorin e_6_ 3(1),3(2)-*bis*-(N,N,N-trimethylaminomethyl iodide)-13(1)-N′- (2-N′′,N′′,N′′-trimethyl ammonioethyl iodide) amide 15(2),17(3)-dimethyl ester (comp. **4**).

**Table 1 ijms-23-05294-t001:** Partition coefficients in the 1-octanol/PSB biphasic system and thermodynamics of solute transfer at 298–318 K.

*T*, K	1	2	3	4 ^2^
	Partition coefficients			
298.15	1.88 ± 0.06 ^1^	20.1 ± 0.9	16.8 ± 0.8	0.97 ± 0.03
308.15	1.90 ± 0.09	22.7 ± 0.8	17.2 ± 0.9	1.04 ± 0.02
318.15	1.91 ± 0.10	24.1 ± 0.7	17.7 ± 0.8	1.11 ± 0.03
Standard free energies (Δ_t_*G*^0^) and enthalpies (Δ_t_*H*^0^) of transfer at 298.15 K				
Δ_t_*G*^0^, kJ mol^−1^	−0.560 ± 0.001	−7.5 ± 0.06	−5.00 ± 0.01	0.08 ± 0.001
Δ_t_*H*^0^, kJ mol^−1^,	0.14 ± 0.03	7.1 ± 1.3	4.00 ± 0.2	5.32 ± 0.003
*r*_f_, kJ mol^−1^	0.003	0.19	0.03	0.001

^1^ This value is taken from ref. [13], ^2^ ref. [28]. The uncertainties represent the standard error.

**Table 2 ijms-23-05294-t002:** Parameters of Equation (3) and the Stern–Volmer constants for comps. **1**–**4**.

Parameter	1	2	3	4 ^1^
	Equation (3)			
*m*_T 1_, mol kg^−1^	(0.23–0.88)·10^−4^	(0.35–4.21)·10^−4^	(0.40–1.75)·10^−3^	(1.6–5.6)·10^−5^
*m*_T 2_, mol kg^−1^	(1.1–7.3)·10^−4^	(0.56–1.12)·10^−3^	(1.75–2.50)·10^−3^	(0.6–2.2)·10^−4^
*n* _1_	0.86 ±0.01	0.45 ± 0.04	1.43 ± 0.13	0.33 ± 0.03
*n* _2_	1.99 ± 0.1	3.58 ± 0.4	7.77 ± 0.7	3.19 ± 0.7
lg *K*_b 1_	3.79 ± 0.03	2.10 ± 0.14	4.37 ± 0.40	1.44 ± 0.13
lg *K*_b 2_	8.39 ± 0.4	12.48 ± 1.2	21.92 ± 1.7	14.1 ± 2.9
	Equation (4)			
*K*_SV_, kg mol^−1^	0.71 ± 0.07 (200) ^2^ 2.62 ± 0.03 (60)	0.73 ± 0.02 (200)	1.25 ± 0.06 (200)	2.39 ± 0.10 (200) 5.40 ± 0.18 (60)

^1^ Ref. [28], ^2^ the values given in parentheses are the Tween 80/PS molar ratio in a solution. The uncertainties represent the standard error.

**Table 3 ijms-23-05294-t003:** Quantum yield of singlet oxygen ^1^O_2_ for comps. **1**–**4** in OctOH, PS accumulation in K-562 cancer cells and PS light toxicity (%).

Control	1	2	3	4
Singlet oxygen quantum yield				
-	0.56 ± 0.03 ^1^	0.63 ± 0.02	0.74 ± 0.02	0.53 ± 0.05 ^1^
PSs accumulation in K-562 cells (integral fluorescence measurements) ^2^				
1.4	23.0	89.2	70.2	-
Percentage of photo-inactivated cells ^2^				
2.1 ^3^	9.1	31.4	22.2	15.9 ^5^
3.46 ^4^	14.2	39.8	39.2	

^1^ Refs. [13,28]. The uncertainties are the standard errors; ^2^ the error of cell experiments was estimated to be within 5–10%; ^3^ 0.44 J cm^−2^; ^4^ 0.66 J cm^−2^; ^5^ this value was obtained for the HeLa cell line with the light dose of 12 J cm^−2^ and *C*_PS_ = 1 μM [4]. The control gives light toxicity without PS.

## Data Availability

The data presented in this study are available on request from the corresponding author.

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
