# Peer review of "Solvation, Cancer Cell Photoinactivation and the Interaction of Chlorin Photosensitizers with a Potential Passive Carrier Non-Ionic Surfactant Tween 80"

_ijms, 2022, doi:10.3390/ijms23105294_

Round 1
Reviewer 1 Report
The paper of Kustov et al is a very interesting study regarding the synthesis and characterization of some photosensitizes that could be combined with Tween 80 in order to improve their effect in photodynamic therapyThe presented results are few, compared to the supplementary files. In my opinion, the NMR spectra should be presented in the paper body. The paper is well written.
Author Response
The Reviewer 1
The paper of Kustov et al is a very interesting study regarding the synthesis and characterization of some photosensitizes that could be combined with Tween 80 in order to improve their effect in photodynamic therapy. The presented results are few, compared to the supplementary files. In my opinion, the NMR spectra should be presented in the paper body. The paper is well written.
Dear the Reviewer,
Thank You very much for the comment made. We have added this and some other information in the main text.
Sincerely yours
Andrey Kustov
Reviewer 2 Report
- Even though this manuscript focused on the partition coefficient of chlrorin derivatives, basical PDT efficacy against cancer cells shpuld be addressed. For example, PDT efficacy at cetain light dose can be applicable to cancer cells (for example, B16F10 cell or HeLa cell) and cell viability can be compared with original chlrorin e6 (Ce6)
- ROS generation of chlorin must be affected by modified structures. And, the main purpose of chlorin for PDT is to generate ROS for killing cancer cells. Then, ROS productivity of each compound also compared with original chlrorin.
- Fluorescence and/or UV spectra of each compound should be addedd and sappropriate wavelength sould be indicated. For example, chlorin e6 (Ce6) is known to be activated at 664 nm and, in reference, Ce6 has maximum peak at 664 nm. I recommend fluorescence or UV spectra of each compound in water (PSB) and/or organic solvent.
- If some compound among compound 1 ~ 4 forms micelles, would you present critical micelle concentration of compound ?
- Please move some results from supplementary materials to main manuscript. For example, Table S1 and S2 should be presented in main manuscript.
Author Response
The Reviewer 2
Dear the Reviewer,
Thank You very much for the comments made. Our answers are given below.
- Even though this manuscript focused on the partition coefficient of chlorin derivatives, basical PDT efficacy against cancer cells should be addressed. For example, PDT efficacy at certain light dose can be applicable to cancer cells (for example, B16F10 cell or HeLa cell) and cell viability can be compared with original chlorin e6 (Ce6)
Thank You for this comment. The available information on photoinactivation of cancer cells is added to the manuscript.
- ROS generation of chlorin must be affected by modified structures. And, the main purpose of chlorin for PDT is to generate ROS for killing cancer cells. Then, ROS productivity of each compound also compared with original chlorin.
Thank You for this comment. The quantum yield values are added to the main text and briefly discussed.
- Fluorescence and/or UV spectra of each compound should be added and appropriate wavelength should be indicated. For example, chlorin e6 (Ce6) is known to be activated at 664 nm and, in reference, Ce6 has maximum peak at 664 nm. I recommend fluorescence or UV spectra of each compound in water (PSB) and/or organic solvent.
Thank You for this comment. This information is added to the Suppl. Mat. File.
- If some compound among compound 1 ~ 4 forms micelles, would you present critical micelle concentration of compound ?
Thank You for this question. In fact, comp. 3 forms aggregates at concentrations of 1 mmol/kg. This concentration is much larger than those studied in our paper (0.01 mmol/kg or smaller). Moreover, the flat structure of macrocycles does not allow them to form classical micelles as it usually takes place for surfactants, for example, Tween 80. The CMC of this carrier is given in the main text.
- Please move some results from supplementary materials to main manuscript. For example, Table S1 and S2 should be presented in main manuscript.
Thank You for this question. We have significantly modified the text and added new information.
Sincerely yours
Andrey Kustov
Round 2
Reviewer 2 Report
They responded adequately.